# A Study on Inter-Provincial Environmental Pollution Movement in China Based on the Input–Output Method

**Yong Shi [1,2], Anda Tang [3] and Tongsheng Yao [4,*]**

1    Research Center on Fictitious Economy and Data Science, Chinese Academy of Sciences, Beijing 100190, China
2    The Key Laboratory of Big Data Mining and Knowledge Management, Chinese Academy of Sciences, Beijing 100190, China
3    School of Mathematical Sciences, University of Chinese Academy of Sciences, Beijing 100190, China
4    School of Economics and Management, University of Chinese Academy Sciences, Beijing 100190, China
*    Correspondence: yaotongsheng20@mails.ucas.edu.cn

**Abstract:** In China, environmental pollution responsibilities are divided according to administrative regions. However, because of the strong externality of environmental pollution, the movement of environmental pollution undoubtedly increases the complexity of pollution governance. To divide the responsibility of environmental pollution governance in each province, we effectively quantify the unequal relationship between environmental pollution costs and economic benefits in each province to understand the mechanism and characteristics of inter-provincial environmental pollution movement. Based on the regional input–output model and an inventory we compiled of sulfur dioxide($SO_2$) emissions of air pollutants in 2012, we calculate the implied inter-provincial environmental pollution emissions and economic benefits from trade, based on production and consumption (supply-side and demand-side). In addition, the movement relationship is explored, and the cost and economic value-added indexes of air pollution control are further constructed to provide effective evidence for a reasonable division of responsibility for environmental pollution control. The results show that there are obvious environmental inequities in the regional trade process in each province, indicating that environmental pollution has been moved. Developed provinces have more economic benefits but pay less in the process of trading goods with less-developed provinces due to the advantages of their industrial structure, while the opposite is true for less-developed provinces. Finally, we propose corresponding policy recommendations to change this condition.

**Keywords:** environmental pollution; input and output; emission inventory



## 1. Introduction

The rapid development of China's economy has inevitably brought about many problems, including environmental pollution Kan et al. [1]. On the one hand, the development of various industries, while employing many people and creating value, inevitably brings about environmental pollution. On the other hand, the large concentration of population and the imbalance of population structure lead to the demands related to an increaesd number of people with limited resources, thus bringing a heavy burden to the ecological environment.

The issue of environmental pollution has long been of concern to scholars. Some scholars use econometric empirical methods (for example, Liu and Lin [2], Dou and Han [3], Lan et al. [4]). Further, there are a handful of studies using IO methods to look at air pollution. Guan et al. [5] measured the magnitudes of socioeconomic factors in driving primary $PM_{2.5}$ emissions changes in China between 1997–2010 by using a regional emissions inventory as input into an environmentally extended input–output framework and applying structural decomposition analysis. Wang et al. [6] tracked the movement of environmental pollution in inter-provincial trade using a multiregional input–output model. Zhu et al. [7] quantified

the provincial energy intensity of regional water pollutant removal by wastewater treatment plants and tracked sectoral and regional water pollutant (COD) movements under provincial economic transactions based on multiregional input–output analysis, and then measureed savings and losses in different provinces based on an energy–pollutant relationship assessment framework. Zhu et al. [8] proposed an inter-country input–output decomposition framework that can distinguish between domestic and multinational firms, and recalculated the $CO_2$ emissions of the global value chain, including international trade, foreign direct investment, and the emissions between these two factors. Yan and Ge [9] constructed regional economic impact coefficients (REIC) and $CO_2$ emission impact coefficients (RCIC) based on the multiregional input–output (MRIO) model and analyzed the economic–carbon relationships of 17 industries in 30 regions in China in 2010. Wang et al. [10] used an environmental input–output model to assess implied water and energy consumption, measuring water and energy consumption coefficients and $CO_2$ emissions factors. Wen and Wang [11] constructed MRIO models for 30 Chinese provinces in 2002, 2007, and 2010 to measure the implied $CO_2$ emissions in China's cross-regional trade at the regional and industry levels, exploring their changes over time and analyzing the driving forces of inter-regional $CO_2$ emissions by final demand through SDA models. Zhang et al. [12] applied the input–output (I-O) framework to study the influencing factors of energy consumption and intensity as well as final demand and industry-level changes in Shanxi Province from 2002 to 2017.

Furthermore, some scholars combine the input–output method with other methods, further expanding the application of the input–output method. Wang et al. [13] integrated input–output modeling (IOM) and two-stage factor analysis (TFA) into a general framework to identify the main factors of $CO_2$ emissions and to quantify $CO_2$ emissions. Huang et al. [14] used a two-level input–output structural decomposition analysis to identify the driving forces of $CO_2$ emissions in countries with energy resource-poor and fossil fuel-centered economies (e.g., Taiwan, Japan, and Korea). Meng and Sager [15] studied the overall situation of energy consumption and energy-related $CO_2$ emissions in China's petrochemical industry in 2012 by using environmental input–output life cycle assessment (EIO-LCA). Li and Li [16] developed a multi-scenario integrated simulation and environmental input–output (MES-EIO) model to simulate multi-pollutant and $CO_2$ emissions under different scenarios over a long-term planning horizon (2020–2030). Duan et al. [17] combined multiregional input–output analysis (MRIO) and ecological network analysis (ENA) to assess carbon emissions in China and to identify critical regions and sectors in the context of spatial heterogeneity to build an inter-regional carbon network model. After that, ENA was used to reveal indirect carbon flows and inter-regional interrelationships. Wang and Song [18] measured the factors responsible for the decline in coal consumption by combining multiregional input–output analysis and the log-mean index method and concluded that the decline in coal consumption in the UK was the result of energy transition and outsourcing. He et al. [19] analyzed net energy consumption in Australia from 2004–2005 to 2014–2015 by decomposing changes in energy consumption into five factors through an environmental extended input–output model and structural decomposition analysis (SDA). Gao et al. [20] investigated the connection between water and particulate matter 2.5 ($PM_{2.5}$) by extending the environmental multiregional input–output model with the integrated Nexus intensity index. Hubacek et al. [21] detailed consumer expenditure surveys for different income categories across a wide range of countries and combined them with an environmentally extended multiregional input–output approach to estimate the global carbon footprint of different household groups, to explore global carbon inequality and the carbon impact of poverty reduction between and within countries, and to assess the carbon impact of lifting the poorest out of poverty.

The actual resource consumption or emissions within a given territory (often a region or country) are referred to as the production side or production perspective of accounting for emissions. Many scholars have carried out research based on this viewpoint. Cheng [22] studied the spatial correlation and interaction between manufacturing agglomeration and environmental pollution by using statistical data from 285 cities in China. Yang et al. [23]

studied the effect of producer services and manufacturing industrial agglomeration on the ecological environment pollution control by using the spatial Durbin model. Ren et al. [24] used a vector error correction model (VECM) and three-stage simultaneous equations (3SLS) to explore the mechanisms between industrial agglomeration and environmental pollution. Zhang et al. [25] proved that increased industrial agglomeration across China has significantly worsened environmental pollution based on joining 112 kinds of spatial econometric model structures and the panel data of 30 provinces in China from 2003 to 2016.

There are more and more disputes and doubts about the definition of the responsibility for pollution reduction based on the principle of producer responsibility. Pollution emissions accounting based on consumer responsibility (measuring resource consumption and pollution emissions of a specific region from the perspective of consumption and demand) has quickly become a hot research direction in the academic field (Proops et al. [26]). Meng et al. [27] adopted consumption-based accounting and identified the original global source that produced the emissions-based MRIO. Moran and Kanemoto [28] linked $SO_2$, $NO_x$, and $PM_{10}$ severe emissions hotspots to final consumers via global supply chains. Nagashima et al. [29] combined an input–output model with an atmospheric transport model and fully linked consumer demand to final pollutant fate and health impact to identify the key sectors responsible for primary carbonaceous $PM_{2.5}$ mortality. Wu et al. [30] calculated the $PM_{2.5}$ emissions embodied in provincial trade (EEPT) of China's 30 provinces in 2007 and 2010 based on a multiregional input–output model.

Because environmental pollution has strong externalities (Koomey and Krause [31]), solving environmental pollution problems is very complicated. For example, the Beijing–Tianjin–Hebei region has different industrial structures , with Beijing dominated by primary industry and Hebei by tertiary industry(Chu et al. [32] Chen and Li [33] Hu et al. [34]). Hebei has highly polluting heavy industries such as metal smelting and metal processing, while Beijing has mainly tertiary industries, which are far less destructive to the environment than Hebei's. However, it is incorrect that Hebei should bear more responsibility for environmental pollution. Although Beijing has closed down highly polluting enterprises in recent years and transferred them to Hebei and other places, the products produced in Hebei are also consumed by Beijing, so from this point of view, it can be said that Beijing has grafted its environmental pollution onto Hebei. However, of course Beijing should not have to take full responsibility for the pollution, because while Beijing was grafting pollution onto Hebei, these highly polluting enterprises were also bringing jobs and economic growth to Hebei. Therefore, the key to solving the problem is to clarify the regional air pollution responsibility-sharing mechanism. Thus, to clarify the responsibility-sharing mechanism of air pollution in the region, we must clarify the mechanism and characteristics of air pollution movement in the region. If a region (province) gains economic benefits from trading goods and services with other regions (provinces) without taking reasonable responsibility for pollution government, this asymmetry and irrationality prevents a fair and reasonable distribution of the "emission reduction" responsibilities that should be borne by provinces and municipalities, thus affecting the synergy of regional air management. This affects the synergistic effect of regional air government.

Therefore, the goal of this study is to characterize inter-provincial environmental pollution movement in China and the interrelationship of economic benefits, and to set up an environmental inequity index to quantify inter-provincial environmental inequality (rationality). In this paper, inter-provincial trade is used to quantify the economic benefits to each region.

Although there is much previous research on the emission and movement of environmental pollutants, most of them singularly account for pollution responsibility. The two main contributions of this study are as follows:

(1) Based on both the MRIO model and $SO_2$ data, there are no $SO_2$ data at the industry level, but only the total $SO_2$ emissions in the 2012 statistical yearbook of 31 provinces. (It should be noted that the environmental pollutants mentioned in this paper re-

fer to $SO_2$, $NO_2$, PM, and other pollutants harmful to the human body. $CO_2$ can cause climate change, but it has no significant effect on human health over a short time. However, $SO_2$, $NO_2$, PM, and so on cause significant harm to the human body (Liang et al. [35] Zhang et al. [36] Zhao et al. [37] Zhao et al. [38]). Further, we have not found relevant third-party data, so we have compiled our own pollution emission inventory). Because of the data that we can obtain, the research object of this study is $SO_2$. In the follow-up study, we will study $NO_x$, $PM$ and other pollutants. This study measures and quantifies the economic benefits and environmental pollution movement from the consumption perspective and production perspective, respectively, and establishes an index to quantify the degree of differentiation between these two that makes it easier to study the characteristics of inter-provincial regional environmental pollution movement and expands the application of MRIO theory and modeling more than previous studies in terms of application. This is the first innovation in this paper.

(2) We change $SO_2$ pollution emissions into a quantitative scale that can be compared in value, that is, the cost of complete environmental pollution treatment, which is conducive to construct of an environmental inequity index that reflects air pollution emissions and economic benefits, thus laying the foundation for the fairness and quantification of inter-provincial environmental pollution movement.

## 2. Data and Methods

Air pollutant emission inventory data and multiregional input–output tables for China are the core data for this study. Air pollutant emission inventory data were estimated by using energy consumption and material balance algorithms, and it should be noted that Hong Kong, Macau, and Taiwan provinces were not included due to lack of data. The material balance method is a conventional method for calculating pollutant emissions. Adeniran et al. [39] carried out a material balance, bottom-up emission factor approach and exergy analysis to estimate pollutant emissions. Nazaroff and Singer [40] evaluated the Inhalation of Hazardous Air Pollutants-based material-balance model. Zhou et al. [41] created an emissions inventory of anthropogenic air pollutant sources with a material-balance model and emission factors.

The final air pollution emission inventory ($SO_2$) of China in 2012 was calculated using the emission factors method based on data from provincial statistical yearbooks, the China Environmental Statistical Yearbook (Editorial Committee and Editorial Staff of China Environmental Statistics Yearbook 2013, China Environmental Statistics Yearbook, China Statistics Press, 2013, p4–p5), and the China Energy Statistical Yearbook (Editorial Board and Editors of China Energy Statistics Yearbook 2013, China Energy Statistics Yearbook, China Statistics Press, 2013, p4–p5) subdivided into 31 provinces and 31 industries. The multiregional input–output table of China was obtained from the team of Liu Weidong (Liu W D, Tang Z P, Han M Y, et al. [42]), a researcher at the Institute of Geographical Sciences and Resources (Chinese Academy of Sciences). Then, the value-based input–output table of 42 sectors in 31 provinces of China in 2012 was used as the base data. The data were processed to finalize the multiregional input–output table of 31 sectors in 31 provinces of China in 2012. The two were combined to establish the environmental multiregional input–output model. Specifically, the air pollutant emission inventory is constructed as follows.

### 2.1. Data

In this paper, the various energy sources are converted into standard coal and thus are accounted for as energy consumption. Based on the availability of data and the fact that most of the industries analyzed produce using industrial coal-fired boilers, the material balance algorithm was chosen to account for pollutant emissions.

The variables that affect the pollutant emissions of $SO_2$ are the amount of sulfur ($S$) contained, the boiler combustion method , and the conversion rate ($1 - \eta$). Differences in the efficiency of desulfurization measures also result in different $SO_2$ emissions factors.

$$SO_2 \quad production : Gso_2 = 2 * S_y * P = 2 * S * P * C \tag{1}$$

$$Emission \quad of \quad SO_2 : Cso_2 = Gso_2 * (1 - \eta) \tag{2}$$

where ($C$) denotes the consumption of energy, ($S$) denotes the sulfur content in coal combustion, ($P$) denotes the conversion rate, and $\eta$ denotes the desulfurization rate.

Differences in geography and coal origin lead to differences in the sulfur content of coal combustion. Generally speaking, it is lower in the northern region than in the southern region. The conversion rate is generally 80–85% by actual measurement. In this study, 80% was taken. (http://www.chinanecc.cn/website/News!view.shtml?id=118381, accessed on 20 May 2022) . Most of the dust removal systems used in the country rely on hemp heart stone centrifugal technology, and the desulfurization efficiency is generally 20–30%.

### 2.1.1. Data Source

Different sulfur combustion in industrial boilers and the resulting sulfur dioxide production and emission factors are shown in the following Tables 1 and 2, respectively.

**Table 1.** $SO_2$ pollution coefficient of coal-fired industrial boilers.

| P(%) | $S_y$ (%) | | | | | | |
|---|---|---|---|---|---|---|---|
| | 0.5 | 1 | 1.5 | 2 | 2.5 | 3 | 3.5 |
| 80 | 8 | 24 | 24 | 32 | 40 | 48 | 56 |
| 85 | 8.5 | 25.5 | 25.5 | 34 | 42.5 | 51 | 59.5 |

**Table 2.** $SO_2$ emission factors of coal-fired industrial boilers at 80% and 85% (P).

| $\eta$ | P (%) | 0.5 | 1 | 1.5 | 2 | 2.5 | 3 | 3.5 |
|---|---|---|---|---|---|---|---|---|
| 10 | 80 | 7.2 | 14.4 | 21.6 | 28.8 | 36 | 43.2 | 50.4 |
| 10 | 85 | 7.65 | 15.3 | 22.95 | 30.6 | 38.25 | 45.9 | 53.55 |
| 20 | 80 | 6.4 | 12.8 | 19.2 | 25.6 | 32 | 38.4 | 4.8 |
| 20 | 85 | 6.8 | 13.6 | 20.4 | 27.2 | 34 | 40.8 | 47.6 |
| 30 | 80 | 5.6 | 11.2 | 16.8 | 22.4 | 28 | 33.6 | 39.2 |
| 30 | 85 | 5.9 | 11.9 | 17.85 | 23.8 | 29.75 | 35.7 | 41.65 |
| 40 | 80 | 4.8 | 9.6 | 14.4 | 19.2 | 24 | 28.8 | 33.6 |
| 40 | 85 | 5.1 | 10.2 | 15.3 | 20.4 | 25.5 | 30.6 | 35.7 |
| 50 | 80 | 4 | 8 | 12 | 16 | 20 | 24 | 28 |
| 50 | 85 | 4.25 | 8.5 | 12.75 | 17 | 21.25 | 25.5 | 29.75 |

Data source: China Energy Strategy Study (2000–2050) Environmental Sub-report (China Energy Strategy Study Group, China Electric Power Press, 1997).

The sulfur content rates (Table 3) determined in this paper are averaged using the corresponding provincial sulfur content rates from other scholars' studies. The conversion rate is generally 80–85% (Zhang Wei, Zhong guo mao yi yin han da qi wu ran zhuan yi yu huan jing bu gong ping yan jiu ("A study on trade implied air pollution movement and environmental inequity in China"), Nanjing University, 2018), and the value taken in this paper is the same as the one in China (2007) No. 36, which is 80% (http://www.chinanecc.cn/website/News!view.shtml?id=118381, accessed on 24 May 2022). The desulfurization efficiency is taken as 20% in this paper. The energy consumption of each province determined in this study is the consumption of each variety of energy in each industry and each province in the corresponding year. The data come from the statistical yearbooks of each province (Jiangsu, Heilongjiang, Zhejiang, Guangxi, and Sichuan statistical yearbooks have the emissions of each pollutant in each industry, which

are directly used without further calculation; the data of Shanghai come from Shanghai Industrial Transportation and Energy Statistical Yearbook, and the data of Zhejiang come from Zhejiang Natural Resources and Environment Statistical Yearbook). (The annual statistical data can be found on the statistical website of each province; for example, the link for coal consumption in Tianjin is: https://stats.tj.gov.cn/nianjian/2013nj/indexch.htm, accessed on 30 August 2022, Zhejiang Data Cited from Editorial Board of Zhejiang Statistical Yearbook of Natural Resources and Environment-2013, Zhejiang Statistical Yearbook of Natural Resources and Environment, China Statistics Press, 2013, p4. In addition, we uploaded the data of Zhejiang and Shanghai in the Supplementary Materials).

**Table 3.** Sulfur content and $SO_2$ emission factors of commercial coal for provinces and cities in China.

| Province | Sulfur Content (%) | Province | Sulfur Content (%) | Province | Sulfur Content (%) |
|---|---|---|---|---|---|
| National | 1.04 | Zhejiang | 1.09 | Hainan | 1.12 |
| Beijing | 0.7433 | Anhui | 1.1433 | Chongqing | 2.32 |
| Tianjin | 0.8433 | Fujian | 1.0333 | Sichuan | 2.1333 |
| Hebei | 0.8667 | Jiangxi | 1.72 | Guizhou | 2.2433 |
| Shanxi | 0.9933 | Shandong | 1.2767 | Yunnan | 2.2233 |
| Inner Mongolia | 1.0867 | Henan | 1.075 | Shaanxi | 1.65 |
| Liaoning | 0.76 | Hubei | 1.0533 | Gansu | 1.72 |
| Jilin | 0.5167 | Hunan | 1.0733 | Qinghai | 0.825 |
| Heilongjiang | 0.5433 | Guangdong | 1.1333 | Ningxia | 1.3967 |
| Shanghai | 0.9733 | Guangxi | 1.93 | Xinjiang | 1.0733 |
| Jiangsu | 1.2567 | | | | |

## 2.1.2. Data Processing

Energy includes fossil energy (such as coal, oil, and natural gas) and non-fossil energy (such as hydropower, nuclear power, wind power, solar energy, and biomass). Fossil energy and biomass contain sulfur components. Biomass is used less in industrial production, so it was not considered in this paper. The main component of natural gas is methane, which has a low sulfur content. Electricity is a secondary energy source; China's electricity is based on hydropower, thermal power, and nuclear power, with only thermal power containing sulfur. It is impossible to distinguish the source of electricity used in each province, so natural gas and electricity are also not considered in this study as sources of $SO_2$. In summary, in the existing energy structure, sulfur-containing energy is mainly coal and fuel oil. We convert coal and fuel oil into standard coal for calculation. The coefficient of conversion of various energy sources into standard coal is shown in Table 4 below. (Note: The total energy consumption by sector in the statistical yearbook is not equal to the total energy consumption by sector (standard coal) because the composition of the total consumption is not only the end consumption by sector but also the amount of losses and process conversion losses shared by each sector (cited from Wu Zhiquan, Zhang Qishan, "Accounting for $SO_2$ Emission Factors per Unit Energy Consumption in Industrial Sub-Industries")).

Due to the lack of data and for other reasons, Shanghai only had coal consumption in 2009 (cited from https://tjj.sh.gov.cn/tjnj/20170629/0014-1000199.html, accessed on 30 August 2020, Editorial Committee of Shanghai Statistical Yearbook 2013, Shanghai Statistical Yearbook, China Statistics Press, 2013, p3). Sichuan only had pollutant data in 2005 (cited from http://tjj.sc.gov.cn/scstjj/c105855/nj.shtml, accessed on 30 August 2020, Editorial Committee of Sichuan Statistical Yearbook 2006, Sichuan Statistical Yearbook, China Statistics Press, 2006, p6–p7). For this type of data, it is assumed that the total ratio of pollution emissions of a particular industry in the province in 2012 is the same as in previous years. However, the assumption is not very accurate because of the lack of data

and the small number of provinces involved, so this assumption can be established and processed. The specific measurement method is as follows:

$$G_i^{r'} = \frac{G^{r'}}{G^r} \times G_i^r \tag{3}$$

where $G_i^{r'}$ denotes the pollutant emissions for industry i of region r in 2012, and $G^{r'}$ denotes the emissions of all pollutants for region r in 2012; $G^r$ and $G_i^r$ denote the total air pollutant emissions for region r and the air pollutant emissions for industry i of region r in previous years, respectively.

**Table 4.** Reference coefficients for various energy sources converted into standard coal.

| Energy Type | Discount Factor for Standard Coal | Unit |
|---|---|---|
| Raw Coal | 0.7143 | kg of standard coal/kg |
| Finely Washed Coal | 0.9 | kg of standard coal/kg |
| Other Washed Coal | | |
| Medium Coal Washing | 0.2857 | kg of standard coal/kg |
| Coal Slurry | 0.2857~0.4286 | kg of standard coal/kg |
| Coke | 0.9714 | kg of standard coal/kg |
| Crude Oil | 1.4286 | kg of standard coal/kg |
| Fuel Oil | 1.4286 | kg of standard coal/kg |
| Gasoline | 1.4714 | kg of standard coal/kg |
| Kerosene | 1.4714 | kg of standard coal/kg |
| Diesel | 1.4571 | kg of standard coal/kg |
| Liquefied Petroleum Gas | 1.7143 | kg of standard coal/kg |
| Refinery Dry Gas | 1.5714 | kg of standard coal/kg |
| Natural Gas | 1.33 | kg standard coal/m$^3$ |
| Coke Oven Gas | 0.5714~0.6143 | kg standard coal/m$^3$ |
| Other Gas | | |
| Incinerator Coal Gas | 0.1786 | kg standard coal/m$^3$ |
| Heavy Oil Catalytic Cracking Gas | 0.6571 | kg standard coal/m$^4$ |
| Heavy Oil Thermal Cracking Gas | 1.2143 | kg standard coal/m$^5$ |
| Coke Gas | 0.5571 | kg standard coal/m$^6$ |
| Pressure Gasification Gas | 0.5143 | kg standard coal/m$^7$ |
| Water Gas | 0.3571 | kg standard coal/m$^8$ |
| Coal Tar | 1.1429 | kg of standard coal/kg |
| Crude Benzene | 1.4286 | kg of standard coal/kg |
| Heat (equivalent) | 0.03412 | kg of standard coal per million joules |
| | 0.14286 | kg of standard coal/1000 kcal |
| Electricity (equivalent) | 0.1229 | kg standard coal/kWh |

Data source: China Energy Statistical Yearbook 2013, Department of Energy Statistics, National Bureau of Statistics, China Statistics Press.

### 2.2. Model

#### 2.2.1. The Basic Model of Multiregional Inputs and Outputs

The American economist Leontief [43] founded and developed the input–output technique . This technique is an applied discipline that finds the interdependence between various production sectors in the economy by studying the quantities produced and consumed by them. After World War II, Leontief used the input–output technique to predict the demand for steel in the United States, the employment situation, the impact of wages on price increases, the impact of increases or decreases in defense spending on the U.S. economy, and so on. With the progress and the depth of the science, the application of input–output techniques began to extend from single economic developments to international trade, energy application, environmental protection, resource utilization, population education, industrial restructuring, and other fields. (Leontief and Strout [44]).

Suppose a country or region is divided into m administrative regions, where r and s, respectively, refer to two random regions. There are n economic departments in each

administrative region, where i and j, respectively, refer to two production departments randomly within the region. Then, according to the row direction balance relationship of the multiregional input–output table (Table 5), we can obtain:

$$
\begin{bmatrix} x^1 \\ x^2 \\ \cdots \\ x^m \end{bmatrix} = \begin{bmatrix} z^{11} & z^{12} & \cdots & z^{1m} \\ z^{21} & z^{22} & \cdots & z^{2m} \\ \cdots & \cdots & \cdots & \cdots \\ z^{m1} & z^{m2} & \cdots & z^{mm} \end{bmatrix} + \begin{bmatrix} y^{11} & y^{12} & \cdots & y^{1m} \\ y^{21} & y^{22} & \cdots & y^{2m} \\ \cdots & \cdots & \cdots & \cdots \\ y^{m1} & y^{m2} & \cdots & y^{mm} \end{bmatrix} + \begin{bmatrix} y^{e1} \\ y^{e2} \\ \cdots \\ y^{e3} \end{bmatrix}
\tag{4}
$$

where $x_i^r$ represents the total output of department i in region r and is the mn×1 column matrix; The factor $z_{ij}^{rs}$ indicates that the j-th department in the s region uses the output of the i-th department in the r region to produce the intermediate input of the products of the industry and is the mn×mn matrix; $y_i^{rs}$ represents the value of the final products produced by department i in region r as the final consumption or use in region s and is the mn×n matrix; $ex_i^r$ represents the value of the final products produced by sector i in region r for export to other countries or regions and is the mn×l column vector. The above formula can be expressed as:

$$
x_i^r = \sum_{s=1}^{m} \sum_{j=1}^{n} z_{ij}^{rs} + \sum_{s}^{m} y_i^{rs} + ex_i^r
\tag{5}
$$

or

$$
x = Ax + y^d + y^e
\tag{6}
$$

$$
x = (I - A)^{-1} \times (y^d + y^e)
\tag{7}
$$

where the vector $x = (x_i^r)$ represents the total output, vector $y^d = (\sum_s y_i^{rs})$ represents the total consumption of domestic final products, vector $y^e = (ex_i^r)$ represents the total export, I is the unit matrix, and $(I - A)^{-1}$ is the Leontief inverse matrix, which represents the value of intermediate products used by sector i in region r to meet the demand of sector j in region s to produce units of final products. In this paper, imports and exports are not considered, so we only use $y^d$.

### 2.2.2. The Basic Model of Environmental Multiregional Inputs and Outputs

The environmental pollution emission intensity coefficient is $f_i^r = e_i^r / x_i^r$, while the column vector of coefficients for all sectors is denoted by $f = (f_i^r)_{mn \times 1}$; the economic value-added coefficient is $d_i^r = v_i^r / x_i^r$, while the column vector of coefficients for all sectors is $d = (d_i^r)_{mn \times 1}$, while $e_i^r$ and $v_i^r$ denote the r region i sector $SO_2$ pollutant emissions and economic value-added, respectively.

$$
E_d = \hat{f} \times (I - A)^{-1} \times y^d
\tag{8}
$$

$$
V_d = \hat{d} \times (I - A)^{-1} \times y^d
\tag{9}
$$

In the above equation, with ˆ representing the diagonal matrix, the domestic final product will be consumed through the chain of the input–output table, bringing economic increases for each province and sector and also making pollutant emissions in the process; the increase in total emissions and the increase in economic value-added are denoted by $E_d$ and $V_d$, respectively.

**Table 5.** The structure of multiregional input–output table.

| | Items | | Intermediate Demand | | | | | | | Final Demand | | | Export | Total Output |
|---|---|---|---|---|---|---|---|---|---|---|---|---|---|---|
| | | | Region 1 | | | ... | | Region m | | Region 1 | ... | Region m | | |
| | | | Department 1 | ... | Department n | ... | Department 1 | ... | Department n | | | | | |
| Intermediate Input | Region 1 | Department 1 | | | | | | | | | | | | |
| | | ... | | | | | | | | | | | | |
| | | Department n | | | $z_{ij}^{rs}$ | | | | | $y_i^{rs}$ | | | $y_i^{re}$ | $x_i^r$ |
| | ... | | | | | | | | | | | | | |
| | Region 1 | Department 1 | | | | | | | | | | | | |
| | | ... | | | | | | | | | | | | |
| | | Department n | | | | | | | | | | | | |
| | Import | | | | $Im_j^s$ | | | | | | | | | |
| Initial Investment | Added Value | | | | $v_j^s$ | | | | | | | | | |
| | Total Investment | | | | $x_j^s$ | | | | | | | | | |

2.2.3. Production-Side and Consumption-Side Air Pollution Emission Accounting

The diagonal matrix of final product consumption is denoted by $\hat{f}_i$, i.e., only the existing region i emission factors in this matrix, and the others are 0.

$$E^{rs} = \hat{f}^r \times (I - A)^{-1} \times y^s \tag{10}$$

$$E^{sr} = \hat{f}^s \times (I - A)^{-1} \times y^r \tag{11}$$

$$E_{net}^{rs} = E^{rs} - E^{sr} \tag{12}$$

By the above formula, we calculated emissions embodied in products that are traded, and developed production- and consumption-based estimates of pollution. Then, the product consumption of region s to all regions leads to the emission of environmental pollutants from region r through the industry chain of the input–output table, and the trade of region s to region r implies the streaming of $SO_2$ pollutant denoted by $E_{rs}$ and vice versa. $E_{net}^{rs}$ denotes the net streaming of air pollutants implied by any two regions; if $E_{net}^{rs} > 0$, it indicates region s emits pollutants and they travel through the atmosphere to region r; if $E_{net}^{rs} < 0$, it indicates region r emits pollutants and they travel through the atmosphere to region s. The net flow of air pollutants from region s to region r is denoted by $E_{net}^{rs}$ and vice versa.

$$E_p^r = \sum_i^m \hat{f}^r \times (I - A)^{-1} \times y^s \tag{13}$$

$$E_c^r = \sum_i^m \hat{f}^s \times (I - A)^{-1} \times y^r \tag{14}$$

where $E_p^r$ denotes the total air pollution emissions in region *r* due to the consumption of products for all regions; $E_c^r$ denotes the total of environmental pollutant emissions in all regions caused by consumption in region *r*.

2.2.4. Production-Side vs. Consumption-Side Economic Benefits Accounting

In the following formula, $\hat{d}_i$ denotes the value-added coefficient of region i. Again, the diagonal matrix contains only the value-added coefficient of region i, and all other coefficients are 0.

$$VA^{rs} = \hat{d}^r \times (I - A)^{-1} \times y^s \tag{15}$$

$$VA^{sr} = \hat{d}^s \times (I - A)^{-1} \times y^r \tag{16}$$

$$VA_{net}^{rs} = VA^{rs} - VA^{sr} \tag{17}$$

Then, $VA^{rs}$ denotes the value-added pull from region s to region r through the input–transmission table industrial chain, which refers to the trade-implied value-added movement from s to r and vice versa. $VA_{net}^{rs}$ denotes the net movement of value-added implied by trade between region r and region s. If $VA_{net}^{rs} > 0$, it indicates a net movement of value-added from region s to region r; if $VA_{net}^{rs} < 0$, it indicates a net movement of air pollutants from region r to region s.

$$VA_p^r = \sum_i^m \hat{d}^r \times (I - A)^{-1} \times y^s \tag{18}$$

$$VA_c^r = \sum_i^m \hat{d}^s \times (I - A)^{-1} \times y^r \tag{19}$$

where $VA_p^r$ denotes the value-added pull to region r from the consumption of products for all regions, and $VA_c^r$ denotes the value-added pull to all regions from the consumption of products in region r.

2.2.5. Regional Environmental Inequity Index (REI Index)

To compare inter-provincial environmental pollution movement, it is necessary to establish an index to evaluate it. In this chapter, the regional environmental inequity index (REI index) is constructed as follows:

$$q^{rs} = \begin{cases} \dfrac{\frac{E^{rs}_{net}}{VA^{rs}_{net}} - m_1}{M_1 - m_1}, & if E^{rs}_{net} > 0 \quad and \quad VA^{rs}_{net} > 0 \\ \dfrac{E^{rs}_{net} - m_2}{M_2 - m_2} + \dfrac{|VA^{rs}_{net}| - m_3}{M_3 - m_3} + 1, & if E^{rs}_{net} > 0 \quad and \quad VA^{rs}_{net} < 0 \end{cases} \tag{20}$$

Theoretically, every two provinces in each chess grid map will have two chess grids, for example, Beijing to Hebei and Hebei to Beijing. The two chess grids have two positive and negative values, and the absolute values are equal. We eliminated negative values for operations. The $|VA^{rs}_{net}|$ indicators are absolute values, indicating the absolute magnitude of the net value-added movement between two regions; $m_1, M_1$ represent. respectively. the minimum and maximum values in $E^{rs}_{net}/VA^{rs}_{net}$; $m_2, M_2$ represent, respectively, the minimum and maximum values in $E^{rs}_{net}$; and $m_3, M_3$ represent, respectively, the minimum and maximum values in $VA^{rs}_{net}$. When the net movement of environmental pollution $E^{rs}_{net}$ and the net movement of value-added $VA^{rs}_{net}$ between two random regions move in the same direction, i.e., both are positive, the ratio of the two $E^{rs}_{net}/VA^{rs}_{net}$ is normalized to some value from 0 to 1. It can be found that when $E^{rs}_{net}$ is larger and $VA^{rs}_{net}$ is smaller, this value is larger, which means that the inequity between regions is larger. When the two variables move in opposite directions in two random regions, i.e., $E^{rs}_{net}$ is positive but the net movement of $VA^{rs}_{net}$ is negative, it means that region r moves pollution to region s while gaining an incremental value from region s instead. Then, the above two net movements are normalized to 0 to 1 and summed. In addition, because the two variables moved in opposite directions, which indicates that the inequity is more serious, 1 is added to the original result to characterize its significance and severity. In summary, the REI index integrally reflects the significance of environmental inequity between inter-provincial regions, with higher values representing greater inequity.

## 3. Results

*3.1. Accounting for Air Pollution and Economic Benefits Based on the Consumption Side and Production Side*

As can be seen (Figure 1), accounting for the consumption side, Shandong, Henan, Inner Mongolia, and Jiangsu are the four provinces that emit the most $SO_2$. The consumption of products and services of all industries in these four provinces created $SO_2$ emissions accounting for a total of 30% of national $SO_2$ emissions. The provinces with the lowest $SO_2$ emissions on the consumption side are Tibet, Hainan, Qinghai, Tianjin, and Beijing, the sum of all of which only accounts for 2.63% of the total $SO_2$ emissions in mainland China and about 23.30% of the $SO_2$ emissions of Shandong, which is the most-emitting province.

The provinces with the highest $SO_2$ emissions for the production side are Shandong, Inner Mongolia, Henan, Tianjin, and Shaanxi, accounting for a total of 43.13% of national $SO_2$ emissions, while the provinces with the lowest $SO_2$ emissions for the production side are Hainan, Tianjin, Shanghai, Beijing, and Qinghai, accounting for a total of 1.83% of national $SO_2$ emissions and only about 16.60% of the largest emissions of Shandong Province. Comparing $SO_2$ emissions from the production side and consumption side of each province, it can be seen that 18 provinces have $SO_2$ emissions from the consumption side that are more than their $SO_2$ emissions from the production side, with the largest differences being in developed provinces such as Shanghai, Zhejiang, Jiangsu, Tianjin, and Beijing, where the $SO_2$ emissions from the consumption side are 2.4 to 4.5 times greater than the production side, indicating that the implied $SO_2$ emissions from all products and services consumed in these provinces are much more than the $SO_2$ emissions implied by local production. In addition, there are 13 provinces where $SO_2$ on the consumption side is smaller than $SO_2$ emissions on the production side, mainly including Hebei, Shanxi, Inner

Mongolia, Shaanxi, Guizhou, and Gansu, whose $SO_2$ emissions on the consumption side are 0.6~0.98 times of $SO_2$ emissions on the production side, indicating that the $SO_2$ emissions implied by all products and services consumed in these energy-rich or less-developed provinces are much smaller than the $SO_2$ emissions implied by local production.

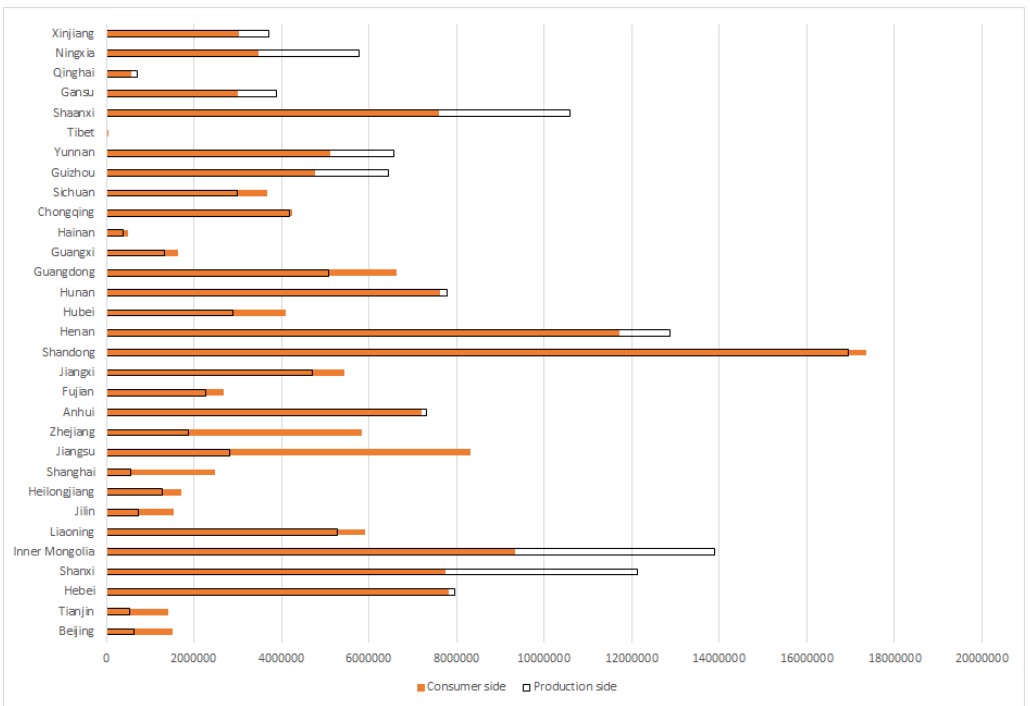

**Figure 1.** Accounting for $SO_2$ emissions from the production and consumption sides in 31 provinces and cities in China.

## 3.2. Type Classification of $SO_2$ and Economic Benefits Net Movement

The 31 provinces in China are divided into four categories based on the net outflow and net inflow of $SO_2$ emissions and economic benefits (Figure 2): provinces in the first quadrant are located in the upper right region and are characterized by net outflow of both $SO_2$ and economic benefits; provinces in the second quadrant are located in the upper left region and are characterized by net inflow of $SO_2$ and net outflow of economic benefits; provinces in the third quadrant are located in the lower left region and are characterized by net inflow of both $SO_2$ and economic benefits; and provinces in the fourth quadrant are located in the lower right region and are characterized by net outflow of $SO_2$ and inflow of economic benefits.

The first quadrant consists of 13 provinces, namely Liaoning, Jilin, Shanghai, Jiangsu, Zhejiang, Fujian, Jiangxi, Shandong, Hubei, Guangdong, Chongqing, Sichuan, and Tibet, which have an outflow of air pollution when consuming other provinces' product, especially highly polluting products, and also drive the economic growth (economic benefits) of the producing provinces in the process of consumption. These provinces are dominated by developed eastern coastal cities, including Shanghai, Jiangsu, Zhejiang, and Guangdong.

The second quadrant consists of three provinces, Anhui, Henan, and Yunnan, which not only receive air pollution in the process of commodity trading with other provinces, but also do not actually receive economic benefits, and they are the areas where environmental inequity is most significantly manifested.

The third quadrant consists of 10 provinces, namely Hebei, Shanxi, Inner Mongolia, Hunan, Guizhou, Shaanxi, Gansu, Qinghai, Ningxia, and Xinjiang, which suffer from the net inflow of air pollution in the process of supplying highly polluting products in the commodity trade with other regions, but receive some economic compensation in the process.

The fourth quadrant consists of five provinces, namely Beijing, Tianjin, Heilongjiang, Guangxi, and Hainan, and these provinces get a win–win situation in the process of commodity exchange with other provinces, sending air pollution to other regions by consuming products from other provinces, and at the same time, throughout the process, the industrial, demographic, and location advantages are used to gain economic benefits from other provinces, causing great inequity to other provinces.

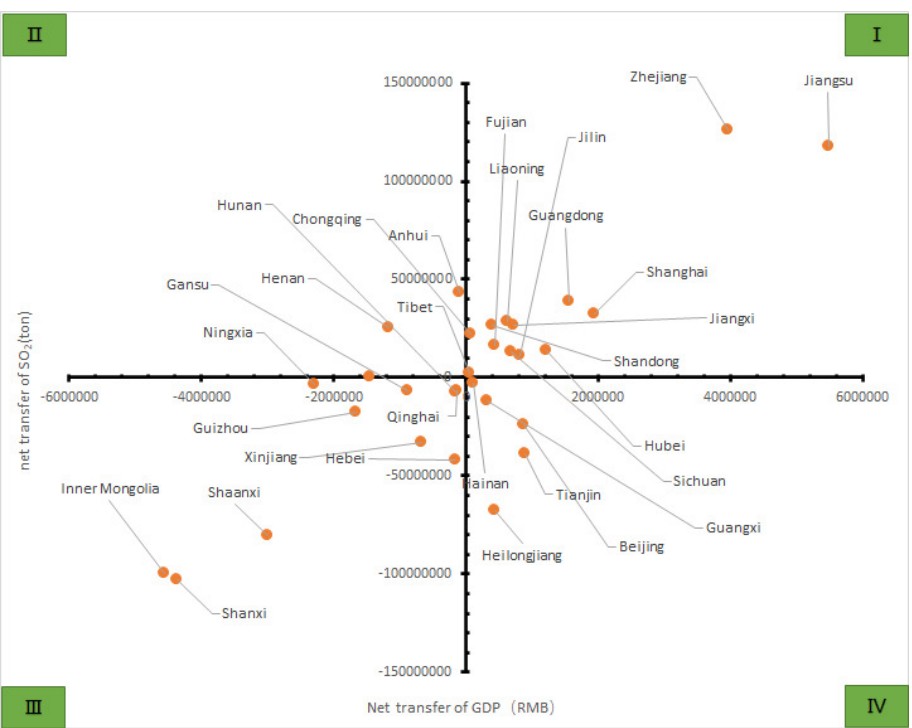

**Figure 2.** Relationship between net outflow of SO$_2$ and net outflow of economic benefits from air pollution in China by provinces.

*3.3. Interprovincial Movement Matrix and Environmental Inequity Index*

3.3.1. Net Interprovincial SO$_2$ Movement

As shown in the Figure 3 below, the provinces of Mongolia, Shanxi, Hebei, Anhui, and Shaanxi are the main provinces with net inflows of SO$_2$ in the process of commodity trading among all provinces, with 4,559,439 tons, 4,375,502 tons, 146,671.7 tons, and 2,986,194 tons, respectively; Jiangsu, Zhejiang, Shanghai, and Beijing are the main provinces with net outflows of SO$_2$ in the process of trading among 31 provinces in China, with 5,481,584 tons, 3,954,425 tons, 1,940,155 tons, and 871,470 tons respectively. Jiangsu has the largest movement of SO$_2$ to other provinces among the 31 provinces. Among them, Jiangsu moved the most to Shanxi, Inner Mongolia, and Shaanxi, with 642,528.9161 tons, 726,931.1735 tons, and 603 884.5549 tons, respectively.

In the surrounding provinces such as Beijing, Tianjin, and Hebei, Beijing and Tianjin transferred more SO$_2$ to the outside, while Hebei, Shanxi, and Inner Mongolia received more SO$_2$. Beijing moved more SO$_2$ to Inner Mongolia, Shanxi, Shandong, and Hebei in the trading process, 150,908.6 tons, 95,311.69 tons, 87,166.92 tons, and 42,652.58 tons, respectively; Tianjin moved more SO$_2$ to Inner Mongolia, Shanxi, Shandong, and Hebei in the trading process, 148,633.8 tons, 121,472.8 tons, 67,903.75 tons, and 44,662.57 tons, respectively. Hebei received 92,312.4 tons, 44,662.6 tons, and 42,652.6 tons from Shandong, Tianjin, and Beijing, respectively. Inner Mongolia, Shanxi, and Hebei, which are rich in coal resources in China, are the main provinces for SO$_2$ inflow. The reason for this is that Inner Mongolia and Shanxi produce a large amount of SO$_2$ from the combustion of fossil fuels (mainly coal) in the process of taking over the "west–east power transmission."

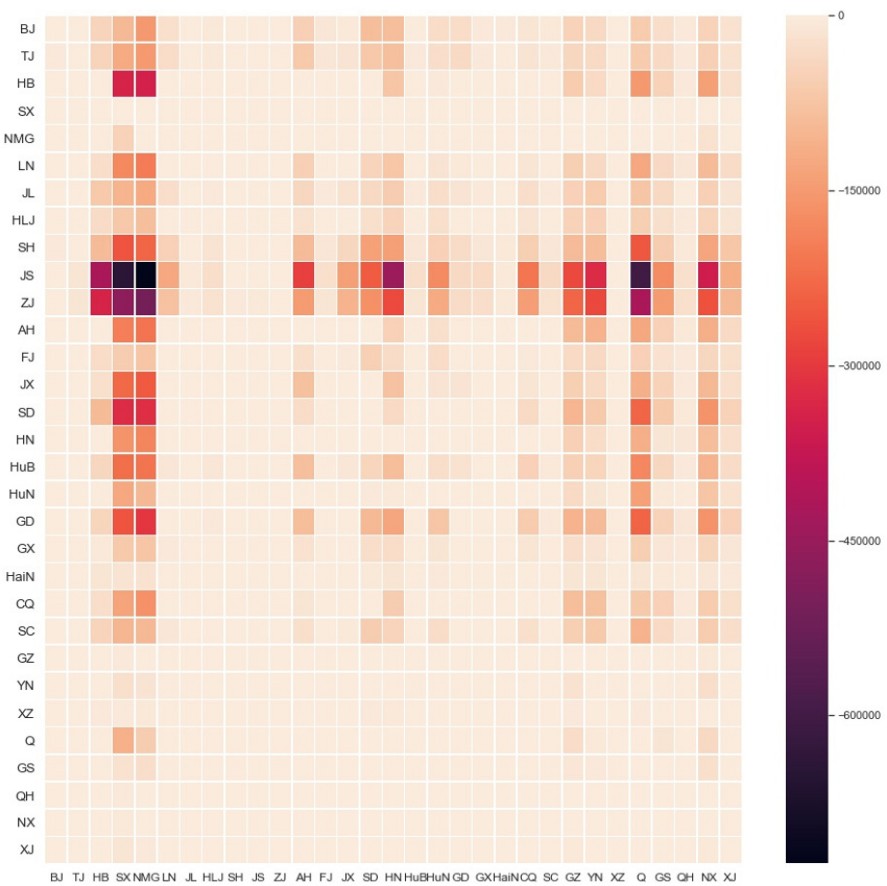

**Figure 3.** Net movement matrix of SO$_2$ (tons) between provinces in China.

### 3.3.2. Net Interprovincial Economic Benefit Movement

As shown in the Figure 4 below, Shanxi, Inner Mongolia, Hebei, Tianjin, and Beijing are the major provinces with net economic benefit inflows in trading out of China's 31 provinces, with CNY 10,285,575,665,000, CNY 9,978,265 964,000, CNY 4,180,566,613,000, CNY 3,853,034,468,000, and CNY 239,805,056,300, respectively. Zhejiang, Jiangsu, Anhui, Shandong, and Henan are the main provinces of China's 31 provinces with net outflows of economic benefits in trading, with CNY 125,871,551 million, CNY 117,417,720 million, CNY 43,481,743.32 million, CNY 26,449,314.5 million, and CNY 25,513,331.19 million, respectively. Zhejiang is the province that moved the most economic benefits to other provinces. Zhejiang moved the most to Hebei, Inner Mongolia, Shanxi, and Shaanxi, moving CNY 1,542,669,592,000, CNY 1,493,235,350, CNY 1,432,615,240,000, and CNY 1,058,034,200, respectively; Shanxi is the province to which other provinces moved the most economic benefits, and Jiangsu, Zhejiang, Guangdong, and Shandong moved the most to it, respectively, CNY 199,793,130,000, CNY 143,236,150,000, CNY 857,070,130,000, and CNY 783,688,000.

Further, out of Beijing, Tianjin, Hebei, and other surrounding provinces, Beijing, Tianjin, Hebei, Inner Mongolia, and Shanxi bear more economic benefit inflows; Shandong moved more economic benefits to the outside. Beijing moved CNY 194,014,140,000 to Tianjin and CNY 146,018,000 to Inner Mongolia, and bore CNY 78,292,911,000 from Shanxi and CNY 295,593,000 from Hebei. Tianjin moved CNY 427,181,000 of economic benefits to Shanxi, and endured CNY 123,485,100, CNY 194,041,400, and CNY 380,358,700 from Inner Mongolia, Beijing, and Hebei, respectively. Hebei moved CNY 757,041,100,000, CNY 61,938,660,000, CNY 380,358,700, and CNY 295,593,000 to Shanxi, Inner Mongolia, Tianjin, and Beijing, respectively.

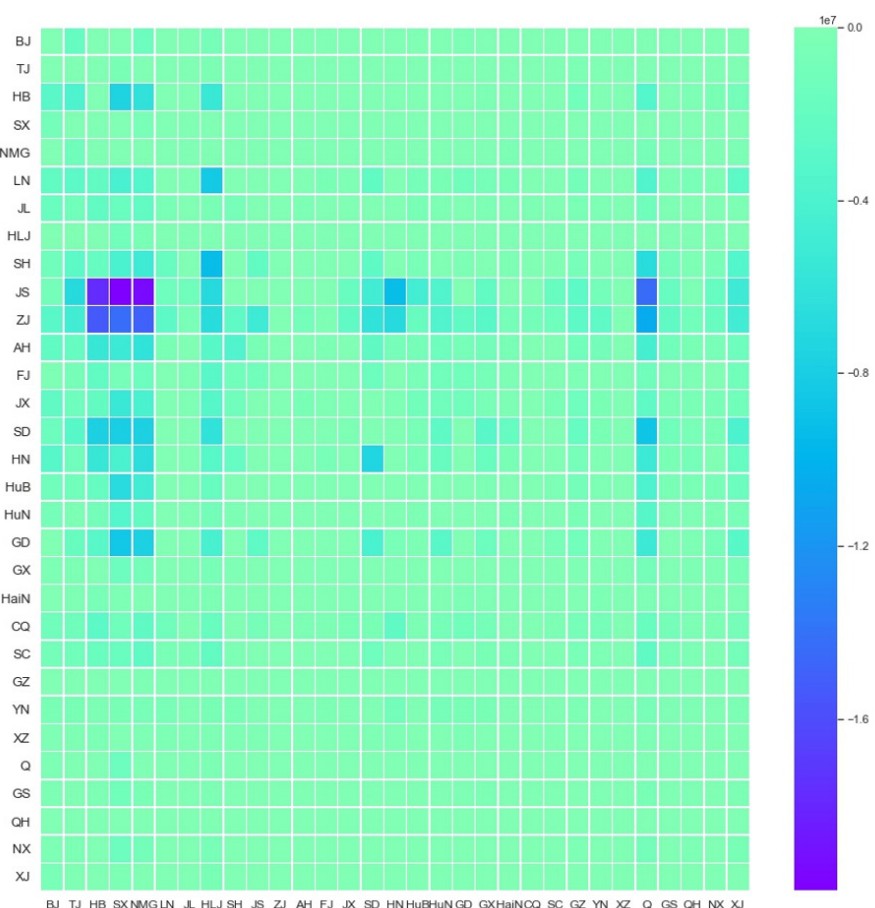

**Figure 4.** China's inter-provincial economic benefits (CNY) net movement matrix.

3.3.3. Interprovincial REI Environmental Inequity Index

From the inter-provincial $SO_2$ and economic benefit movement variables, an index is constructed to obtain the inter-provincial environmental inequity REI index (see the Figure 5 below). This REI index represents the significant degree of relative inequity. In calculating the REI index, we divided all provinces into two types: one has opposite directions of $SO_2$ and economic benefit flows, such as between Beijing and Hebei provinces, where Beijing moved 42,652.58489 tons of $SO_2$ emissions to Hebei in the trading process but gained CNY 29 559.304 million of economic benefits from Hebei; the other has the same direction of $SO_2$ and economic benefit flows, such as between Beijing and Inner Mongolia, where Beijing moved 150,908.5639 tons of $SO_2$ emissions to Inner Mongolia during the trading process and also compensated Inner Mongolia with CNY 146,018.092 million of economic benefits.

For the first category of provinces, the results show that the highest REI index values occurred in Jiangsu–Hebei (REI = 2.38), Zhejiang–Hebei (REI = 2.27), Jiangsu–Shanxi (REI = 2.5). For the second category of provinces, the higher ones were Tianjin–Shaanxi (REI = 1), Liaoning–Henan (REI = 0.86), Sichuan–Ningxia (REI = 0.75 ), and so on.

In the Beijing–Tianjin–Hebei metropolitan area, the REI indices of Beijing with other provinces (Tianjin, Hebei, Shanxi, Inner Mongolia, Shandong, and Henan) were 1.60, 1.68, 1.60, 1.57, 1.63, and 1.72, respectively; the REI indices of Tianjin with other provinces (Hebei, Shanxi, Inner Mongolia, Shandong, and Henan) were 1.72, 1.52, 1.66, 1.70, and 1.62, respectively; the REI indices of Hebei with other provinces (Shanxi, Inner Mongolia, Shandong, and Henan) were 1.88, 1.81, 1.89, and 1.83, respectively. It can be seen that there is a significant environmental inequity within the Beijing–Tianjin–Hebei metropolitan area, i.e., there is a strong movement of environmental pollution.

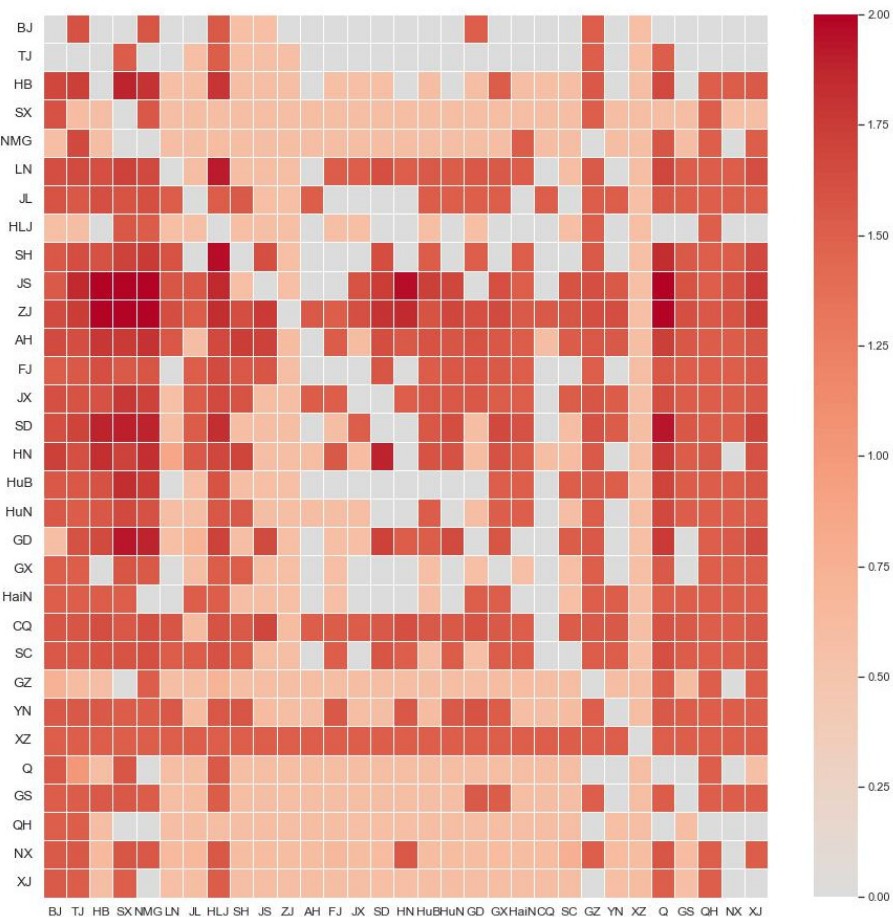

**Figure 5.** China inter-provincial REI index matrix.

## 4. Conclusions

There are significant environmental inequities in the process of commodity trading between provinces in China. Movement outflows mainly happen in developed regions such as the Beijing–Tianjin–Hebei area, the Yangtze River Delta, and the Pearl River Delta, while the inflows are mainly resource- and energy-rich but less-developed provinces such as Inner Mongolia, Shanxi, Hebei, and Henan. The analysis results show that developed eastern coastal regions such as Beijing, Tianjin, Jiangsu, and Shanghai, due to factors such as geographical location, industrial structure, and first-mover advantage, move $SO_2$ out in the process of consuming products and services from other provinces while gaining additional net economic benefit inflows. The less-developed provinces, on the other hand, undertake the movement of $SO_2$ from developed regions in the process of commodity trading, due to a variety of reasons; they do not receive sufficient economic benefit compensation in the whole process but lose economic benefits instead. Further, inter-regional environmental inequity mainly occurs between developed provinces (such as Beijing–Tianjin–Hebei, Jiangsu, Zhejiang, and Guangdong) and less-developed provinces in central and western China (such as Shanxi, Inner Mongolia, and Henan).

Among regional provinces such as Beijing, Tianjin, and Hebei, in relation to Beijing and Tianjin, provinces such as Hebei and Shanxi have gained some economic compensation while bearing the movement of $SO_2$ pollution in the process of trade transactions. However, the REI environmental inequity index is still significant, and an essential factor is the different industrial levels and structures of provinces in Beijing, Tianjin, and Hebei. For example, Hebei, as an agricultural, energy, and resource exporting province, has an industrial structure with primary and secondary industries and can only bear the movement of environmental pollution from developed regions (such as Beijing) with a high level of tertiary industries in the commodity trading process.

The roles and functions of provinces in the Beijing–Tianjin–Hebei region need to be redefined, and the industrial structure needs to be transformed. For example, in Beijing, Tianjin, Hebei, and other provinces, Beijing should make use of political, cultural, international, and other central platforms to develop scientific and technological innovations, continue to vigorously develop tertiary industry, and promote the high-end service industry to integrate with the international community; Tianjin should make full use of its advantages with important ports and the Binhai New Area, and respond to the "Belt and Road" strategy to develop shipping, modern logistics, finance, financial leasing, and other industries, and extend the industrial chain horizontally and vertically. As the top priority of environmental pollution and collaborative governance, Hebei should give full play to its energy resource advantages, vigorously develop modern technology agriculture and eco-tourism, gradually break the restrictions on economic development by high-pollution and labor-intensive industries such as the steel industry, develop clean energy and technology, vigorously develop industries in the new energy field (such as semiconductors, photovoltaics, and new energy vehicles) and high-end manufacturing industries such as electronic information, and complete the transformation and upgrading of the manufacturing industry by relying on the "made in China 2025" plan. In addition, the establishment and development of the Xiongan New Area also laid a foundation for Hebei to establish an economic core point, which has a profound driving effect on the transformation and upgrading of the province's industries, and provides a new way to coordinate and balance economic development and environmental governance.

**Supplementary Materials:** The following supporting information can be downloaded at: https://www.mdpi.com/article/10.3390/en15186782/s1, Table S1: Data of Shanghai and Zhejiang; Table S2: Input and Output Table of 31 Provinces in China in 2012 (42 Departments).

**Author Contributions:** Conceptualization, Y.S.; Data curation, T.Y.; Formal analysis, T.Y.; Funding acquisition, Y.S.; Investigation, T.Y.; Methodology, T.Y.; Project administration, A.T.; Resources Y.S.; Software, T.Y.; Supervision, Y.S.; Validation T.Y.; Visualization, T.Y.; Writing—original draft, T.Y.; Writing—review and editing, A.T. All authors have given approval to the final version of the manuscript.

**Funding:** This work was supported in part by the National Natural Science Foundation of China under grant #72231010 and #71932008.

**Institutional Review Board Statement:** Not applicable.

**Informed Consent Statement:** Not applicable.

**Data Availability Statement:** Not applicable.

**Conflicts of Interest:** The authors declare no conflict of interest.

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
