# Peer review of "A Study on Inter-Provincial Environmental Pollution Movement in China Based on the Input–Output Method"

_energies, doi:10.3390/en15186782_

Round 1

Reviewer 1 Report

Overall comments:

Goal is to attribute responsibility to air pollution by region in China. Nice approach to model material balance from bottom-up data on coal-fired power plants. Yet too many problems throughout to recommend publication.

Not clear how much of total pollution is covered by coal boilers represents and how this compares with other datasets/studies doing IO of air pollution in China.

There is no discussion of physical movement of air pollution between regions, which is important for pollutants that move in the atmosphere (region r burns coal for product in region s, but the pollution actually travels to region z). Many existing studies in the IO literature do this, and this study should at least acknowledge them and discuss why they don’t utilize such an approach.

The MRIO method itself is not described sufficiently, or even possibility accurately, and GDP is conflated with value added throughout the text.

Pieces of input data are missing/unclear, and more data (other input data, regional IO tables) should be provided for scientific reproducibility of estimates.

Grammatical and typological errors throughout.

Find specific comments and suggestions below.

Specific comments:

Page 1

-       Line 2: Instead of “transfer of environmental pollution” maybe say “movement of environmental pollution, which is more accurate and less confusing with a policy mechanism (which I don’t think you mean).

-       Line 9: Economic benefits are not GDP (final demand), but value added (as you later describe)

-       Line 13: Would be nice to have a sentence or two summarizing your main findings and takeaways in the abstract, rather than just a broad assertion that inequities exist (which is obvious).

Page 2

-       Curious to see no citations in the introduction, especially relating to specific cases of Hebei and Beijing where sources for these assertions should be cited.

-       Line 18: The pollutants you consider should be stated later in the intro (not the first paragraph), and given an explanation why you focus on those. And it appears this study only considers S02, not NO2 or particulates.

-       Line 46: Punctuation error.

-       Line 51 – 55: These “issues” are broad questions, which could be more usefully modified into specific research questions. Like “the goal of this study is to define what is an equitable relationship of environmental pollution transfer and economic benefits between Chinese provinces and to test in which cases such equity exists between them.”

-       Related work: This section should be merged with the introduction and to illustrate who has looked at this problem and with what methods. Then you can suggest the approach this study takes and why it is new/applicable, which can then be further explained in the methods. The relevance of some of these studies isn’t immediately obvious and would benefit from this structure for context.

-       Lines 56 – 124: The presentation of studies is not organized very logically. Instead of simply providing a sentence for each relevant study, please organize into type of literature and present in groups to illustrate the different types of approached currently used.

-       Sources 2,3,6,11,16,17,18,19,20,24: Why provide a CO2 and/or water studies when this study does not address it?

-       Source 11: Is the waste water study included to show a previous method for measuring inter-regional savings and losses based on pollutant transfer (even thought not air pollution?)

Page 3

-       Line 91-92: I-O framework is not descriptive, do they use an IO model (Leontief inverse) in some way, or simply use the data?

-       Source 21,22,23: Same for energy/energy efficiency, do you look at this in your study?

-       Line 125-126: Where are the citations for the previous studies using IO and the types of air pollutant transfer you consider (SO2, NO2, PM)? There are plenty, start here:

o   Guan, Dabo, Xin Su, Qiang Zhang, Glen P. Peters, Zhu Liu, Yu Lei, and Kebin He. 2014. “The Socioeconomic Drivers of China’s Primary PM 2.5 Emissions.” Environmental Research Letters 9 (2):24010. https://doi.org/10.1088/1748-9326/9/2/024010.

o   Liang, Sai, Katerina S. Stylianou, Olivier Jolliet, Sarang Supekar, Shen Qu, Steven J. Skerlos, and Ming Xu. 2017. “Consumption-Based Human Health Impacts of Primary PM2.5: The Hidden Burden of International Trade.” Journal of Cleaner Production 167 (Supplement C):133–39. https://doi.org/10.1016/j.jclepro.2017.08.139.

o   Meng, Jing, Junfeng Liu, Yuan Xu, Dabo Guan, Zhu Liu, Ye Huang, and Shu Tao. 2016. “Globalization and Pollution: Tele-Connecting Local Primary PM2.5 Emissions to Global Consumption.” Proc. R. Soc. A 472 (2195):20160380. https://doi.org/10.1098/rspa.2016.0380.

o   Meng, Jing, Junfeng Liu, Yuan Xu, and Shu Tao. 2015. “Tracing Primary PM 2.5 Emissions via Chinese Supply Chains.” Environmental Research Letters 10 (5):54005. https://doi.org/10.1088/1748-9326/10/5/054005.

o   Moran, Daniel, and Keiichiro Kanemoto. 2016. “Tracing Global Supply Chains to Air Pollution Hotspots.” Environmental Research Letters 11 (9):94017. https://doi.org/10.1088/1748-9326/11/9/094017.

o   Nagashima, Fumiya, Shigemi Kagawa, Sangwon Suh, Keisuke Nansai, and Daniel Moran. 2017. “Identifying Critical Supply Chain Paths and Key Sectors for Mitigating Primary Carbonaceous PM2.5 Mortality in Asia.” Economic Systems Research 29 (1):105–23. https://doi.org/10.1080/09535314.2016.1266992.

o   Wu, Leying, Zhangqi Zhong, Changxin Liu, and Zheng Wang. 2017. “Examining PM2.5 Emissions Embodied in China’s Supply Chain Using a Multiregional Input-Output Analysis.” Sustainability 9 (5):727. https://doi.org/10.3390/su9050727.

o   Zhang, Qiang, Xujia Jiang, Dan Tong, Steven J. Davis, Hongyan Zhao, Guannan Geng, Tong Feng, et al. 2017. “Transboundary Health Impacts of Transported Global Air Pollution and International Trade.” Nature 543 (7647):nature21712. https://doi.org/10.1038/nature21712.

o   Zhao, H., X. Li, Q. Zhang, X. Jiang, J. Lin, G. P. Peters, M. Li, et al. 2017. “Effects of Atmospheric Transport and Trade on Air Pollution Mortality in China.” Atmos. Chem. Phys. 17 (17):10367–81. https://doi.org/10.5194/acp-17-10367-2017.

o   Zhao, H. Y., Q. Zhang, D. B. Guan, S. J. Davis, Z. Liu, H. Huo, J. T. Lin, W. D. Liu, and K. B. He. 2015. “Assessment of China’s Virtual Air Pollution Transport Embodied in Trade by Using a Consumption-Based Emission Inventory.” Atmospheric Chemistry and Physics 15 (10):5443–56. https://doi.org/10.5194/acp-15-5443-2015.

-       Line 130: What is “establishes a finger to quantify”.

-       Line 133: I’m not convinced you are advancing MRIO theory, seems like a pretty standard MRIO/EEIO analysis. You would have to be more specific about how what you do is different compared the literature listed above.

-       Line 136: Regarding pricing the externality, or “the cost of complete environmental pollution treatment,” can you please provide a few sources on this as well? This has been done in the U.S. in the 1990s, for instance with a SO2 market and technology for scrubbers (https://www.hks.harvard.edu/publications/us-sulphur-dioxide-cap-and-trade-programme-and-lessons-climate-policy).

-       Line 136: Combining this pricing with IO might be new, but again, please be clear how it is different than IO and air pollution previous (not CO2) studies (for instance, I think it is more common to use statistical value of a human life to monetize externalities).

Page 4

-       Inventory and Methods: should be called methods and data, and then you can explain the “inventory” as part of the data needed for the methods employed.

-       Lines 141 – 150: Paragraph is not needed, very general and doesn’t follow from the detailed lit review you just completed on MRIO and pollution studies.

-       Line 152 – 164: Please explain why you undertook your own air pollution inventory instead of using data from previous studies? Also please cite your sources, such as the China Energy Statistical Yearbook.

-       Line 153: You don’t mention material balance for measuring air pollution emissions until this point, should be mentioned in lit review with any relevant studies.

-       Lines 170 -175: Citations for equations 1-2, or are these your original equations?

Pages 5 - 6

-       No need to include Tables 3 – 5, just cite them an include Table 6.

Page 7

-       Line 185 – 187: Explain why you use the average (are these three different estimates from three studies?).

-       Line 189 – 196: Again, cite yearbooks, this physical consumption data is important.

-       Line 201 typo

-       Lines 197 – 210: An explanation of why you only focus on specific types of energy is nice, but please provide estimates of these percentages (i.e. how much fuel comes from biomass, how much sulfur is in natural gas compared to coal). Then you are more likely to convince readers you are accounting for the majority of SO2 emissions through coal and fuel oil.

-       Line 207: Where are the data and sulfur assumptions for fuel oil? You just said this was included but provide no such method.

-       Lines 211 – 212: Please link to previous (missing) citations of the consumption data in the specific yearbooks where this data is absent.

-       Line 214: You say "although the assumption is not very reasonable”, then why to you use it? Maybe use a different assumption (like tying per capita coal production to other regional/national trends)?

Page 8

-       Table 7 is not even referenced in the paper. Not sure why it is there. Even if referenced, should be in supplementary info, not main paper.

-       Line 223: Is this an MRIO? So (I-A)-1 includes all A matrices from all regions? Please say something to this extent in the methods, you mention MRIO and interregional trade in the data but not the methods.

Page 9

-       You don’t explain the exogenous variable y explicitly anywhere in this section. Of course its final demand, but it’s especially important especially for your production/consumption based estimates that are based on the subset of final demand (imports and exports) moving between regions.

Line 235: f isn’t final consumption, but emissions coefficients, correct? So what you mean to day is “The diagonal matrix of final product consumption in region s (ys) associated with emissions in region r (fr) is calculated using equation 6; since the other values in the vectors y and f are zeros, the remainder of region s consumption and region r emissions are zero.

-       Line 237: Typo: use lower case S for region s

-       Line 242: Please clarify that what you mean is net flow of emissions “embodied in products and services”. This could be confused with the physical flow of pollutants, which much of the literature considers as well (i.e. region s emits pollutants and they travel through the atmosphere to region r)

-       Equation 9 and 10: be sure to specify you mean all regions s from regions i to m. Or does the summation include summing across all r and s (all sets of bilateral trade)?

-       Lines 245-246: Don’t you mean to say: “Erc denotes the total environmental pollutant emissions in all 245 regions due to production in region r.”

-       Section 3.2.3: Not needed at all since it’s simply a direct copy of section 3.2.2, with the exception that VA replaces E. Simply use a sentence to say the same method can be applied to calculate production/consumption side VA.

Pag 10

-       Line 263: Not clear what “the minimum and maximum values in Ersnet/VArsnet” are. Do you mean the minimum and maximum of these values across regions, since you are normalizing across regions? Same for all m and M values. Please make this clearer in the text.

-       No need to list all the detailed numbers, just percentages and totals to illustrate would me more impactful and make for easier reading and takeaways.

-       Line 286: typo

-       Equation 16: what about when net emission transfer is negative? This clearly happens in a bilateral trade figure like Figure 5, which equation did you use in this case?

Page 11

-       Figure 1 would be better with bars for consumption and production are side by side. Easier to see and no confusion about additivity.

-       Section 4.2: you discuss value added in the methods but switch to GDP in this section. Why not keep value added throughout for consistency?

-       Line 317: typo

Page 12

-       Figure 2: maybe highlight or note which quadrants suggest “good” regions and which “bad” (i.e. negative transfer of GDP and positive transfer of S02 good)? Then the reader spends less time figuring out the axes. Also, number quadrants to reflect numbers used in discussion (quadrants 1-4).

Page 13

-       Figure 3: Add units to legend, and specify import and export regions on axes. Also, shouldn’t there be positive and negative values?

-       Again the conflation of GDP and value add, please fix

Page 14

-       Figure 4: again shouldn’t there be positive and negative values (reciprocal values for bilateral trade relationships)

Page 15

-       Figure 5: Change color scale so high values stand out more, right now green stands out more than pink and white. And again, what do you do in the case of negative emission transfer?

Author Response

We gratefully thank the editor and reviewers-for their time-spend-making-their constructive remarks and useful suggestions, which has significantly-raised the quality-of the-manuscript-and has-enable-us to improve-the manuscript. Each' suggested revision and comment, brought forward-by the-reviewers was accurately -incorporated and considered. Below the comments of the reviewers are response point by point and there visions are indicated.

Reviewer 2 Report

The work is interesting and illustrates significant results, however it could be improved in some of its parts. Some suggestions:

The introduction could be widened by including references that allow to give scientific relevance to the topic addressed;

The introduction should include the objective of the work and a paragraph explaining the structure of the work;

The contributions of the work could be moved to the introduction;

If possible, it would be more appropriate to use more recent data for the analysis;

Ensure that all figures are cited and described in the text;

The conclusions could be improved by including the results obtained in the paper, as well as the main limitations and recommendations for further studies.

Author Response

(The authors gave the same response as above.)

Reviewer 3 Report

The treated article deserves a high rating for the specialization of the subject.

The methodological approach meets the minimum requirements usually established and the results obtained are of special interest to science.

The bibliography is extensive and sufficiently related.

A revision in the grammatical forms is suggested.

I resolve to positively qualify the article presented.

Author Response

We gratefully thank the editor and all reviewers for their time-spend-making-their constructive remarks and useful suggestions, which has significantly-raised the quality-of the-manuscript-and has-enable-us to improve-the manuscript. Each' suggested revision and comment, brought forward-by the-reviewers was accurately -incorporated and considered. Below the comments of the reviewers are response point by point and there visions are indicated.

Round 2

Reviewer 1 Report

See additional comments to your responses, in blue text.

Author Response

We gratefully thank the editor and reviewers-for their time-spend-making-their constructive remarks and useful suggestions, which has significantly-raised the quality-of the-manuscript-and has-enable-us to improve-the manuscript. Each' suggested revision and comment, brought forward-by the-reviewers was accurately -incorporated and considered. Below the comments of the reviewers are response point by point and there visions are indicated.

For convenience to read, we have marked the first revision in blue with the number "1" next to it, and the second revision in red with the number "2" next to it.

Reviewer 2 Report

Thanks to the authors for implementing the suggestions, the work is improved in all parts. I recommend a minor revision of the language and style before publication to facilitate reading of the work.

Author Response

(The authors gave the same response as above.)
